# DETERMINE-THEN-ENSEMBLE: NECESSITY OF TOP-k UNION FOR LARGE LANGUAGE MODEL ENSEMBLING

**Yuxuan Yao**[1,2]**, Han Wu**[3†]**, Mingyang Liu**[1,2]**, Sichun Luo**[1,2]**, Xiongwei Han**[3]**, Jie Liu**[4]
**Zhijiang Guo**[5]**, Linqi Song**[1,2†]
[1]Department of Computer Science, City University of Hong Kong
[2]City University of Hong Kong Shenzhen Research Institute
[3]Huawei Noah's Ark Lab
[4]North China University of Technology
[5]Hong Kong University of Science and Technology (Guangzhou)
`yuxuanyao3-c@my.cityu.edu.hk`
`wu.han1@huawei.com`
`linqi.song@cityu.edu.hk`

## ABSTRACT

Large language models (LLMs) exhibit varying strengths and weaknesses across different tasks, prompting recent studies to explore the benefits of ensembling models to leverage their complementary advantages. However, existing LLM ensembling methods often overlook model compatibility and struggle with inefficient alignment of probabilities across the entire vocabulary. In this study, we empirically investigate the factors influencing ensemble performance, identifying model performance, vocabulary size, and response style as key determinants, revealing that compatibility among models is essential for effective ensembling. This analysis leads to the development of a simple yet effective model selection strategy that identifies compatible models. Additionally, we introduce the UNION Top-$k$ Ensembling (UNITE), a novel approach that efficiently combines models by focusing on the union of the top-k tokens from each model, thereby avoiding the need for full vocabulary alignment and reducing computational overhead. Extensive evaluations across multiple benchmarks demonstrate that UNITE significantly enhances performance compared to existing methods, offering a more efficient framework for LLM ensembling. The code is available at `https://github.com/starrYYxuan/UniTE`

## 1 INTRODUCTION

Large language models (LLMs) have demonstrated remarkable performance across a wide range of tasks and have shown promising results in real-world applications (OpenAI, 2023; Yang et al., 2024; Dubey et al., 2024). Given the diversity in data sources, model architectures, and training methods, LLMs exhibit varying strengths and weaknesses depending on the task at hand. Consequently, rather than relying solely on training an LLM from scratch, an alternative approach is to create an ensemble of LLMs. This method allows for leveraging the complementary advantages of different LLMs (Jiang et al., 2023b; Lu et al., 2024; Yu et al., 2024b).

Existing model ensembling methods can be broadly categorized into three types: output-level, probability-level, and training-level approaches. Output-level methods (Jiang et al., 2023b; Lu et al., 2024; Shnitzer et al., 2023) aggregate the complete outputs of multiple candidate models. Probability-level methods (Huang et al., 2024; Yu et al., 2024b), integrate outputs based on probability distributions at each generation step through the intersection or union of the vocabulary. Training-level methods (Wan et al., 2024a; Xu et al., 2024) utilize output probability vectors as labels for richer information extraction during training. While output-level methods are constrained by the limitations of existing outputs, and training-level methods introduce additional computational overhead, probability-level methods have garnered particular attention.

---

† Corresponding authors.

Existing methods for ensembling LLMs, such as DEEPEN (Huang et al., 2024) and GAC (Yu et al., 2024b), grapple with two significant challenges. First, these approaches concentrate solely on the ensembling technique, sidestepping the crucial discussion of which types of models can be effectively combined. This oversight is critical because LLMs with substantial differences in architecture, size, and tokenizer may be inherently incompatible, leading to potential incompatibilities that undermine the benefits of ensembling(Dong et al., 2023; Lee et al., 2023). Second, these methods tend to align the probabilities across the entire vocabulary at each generation step. Such a strategy introduces substantial computational overhead during inference, which hinders performance and efficiency.

To address these challenges, we conduct a thorough investigation into the factors that truly influence ensembling performance. Our empirical analysis identifies three key factors: model performance, vocabulary size, and response process. We evaluate multiple widely recognized LLMs across various benchmarks to explore ensembling from these perspectives. Our findings reveal several important insights: 1) Variations in performance levels among base LLMs significantly affect their compatibility for ensembling; 2) The impact of vocabulary size is marginal; and 3) Even when performance and vocabulary size are aligned across LLMs, substantial differences in the reasoning process in the response can hinder successful ensembling.

Building on this analysis, we adopt a determine-then-ensemble strategy by starting with the best-performing LLM for target tasks, then iteratively selecting the next best-performing LLMs that meets ensembling criteria until the maximum number of models is reached or no further suitable candidates exist. Additionally, we seek to improve the efficiency and performance of ensembling. Instead of aligning the entire vocabulary of different LLMs, we propose the **Uni**on **T**op-$k$ **E**nsembling (**UNITE**). UNITE constructs a union of the top-k tokens from each model and expands this set using each model's tokenizer. This is followed by probability aggregation to determine the next token. UNITE avoids the need for auxiliary mapping matrices and full vocabulary alignment, respecting the unique tokenization of each base LLM. Extensive experiments demonstrate the effectiveness of UNITE across various tasks, consistently outperforming state-of-the-art LLM ensemble methods.

Overall, our key contributions are: 1) We conduct an extensive analysis of existing model ensembling approaches and derive three significant insights. Building on these findings, we introduce a general model selection strategy for model ensembling; 2) We propose a well-motivated model ensembling method that efficiently operates on the top-k candidate tokens rather than the entire vocabulary. 3) Experiments demonstrate significant performance improvements, reduced operational tokens to less than **0.04%** of current methods, and minimized latency to only 10 *ms* longer than the individual model, validating the effectiveness of our proposed approach.

## 2 RELATED WORKS

Based on the sequence of inference and fusion processes, multiple model collaboration methods can be broadly classified into two categories: model ensembling (Jiang et al., 2023b; Yu et al., 2024b) and model merging (Yu et al., 2024a; Akiba et al., 2024). Model ensembling follows an inference-then-fusion approach, aiming to integrate the outputs of various models to achieve a more refined response. Conversely, model merging adopts a fusion-then-inference strategy, wherein different models are combined into a single model before inference. While model merging is typically applicable only to homologous models, our focus in this study is on the more general approach, namely model ensembling. We discuss different model ensembling methods as follows:

**Output-Level Model Ensembling** methods involve selecting multiple candidate models and utilizing their complete outputs for aggregation. For instance, Jiang et al. (2023b) developed PAIR-RANKER, an additional ranking model, to evaluate and select the best candidate output. Similarly, Lu et al. (2024) and Shnitzer et al. (2023) designed a router that determines the most appropriate candidate model based on the given question. However, these approaches are restricted by the existing outputs and become ineffective if all options are incorrect. Some studies have addressed this by training fusion models to combine outputs (Jiang et al., 2023b; Wang et al., 2024b), which alleviates the limitation of relying solely on available candidates and often yields improved results. Nevertheless, achieving generalization with fusion models remains a significant challenge, as they may not fully utilize the probability information generated at each step.

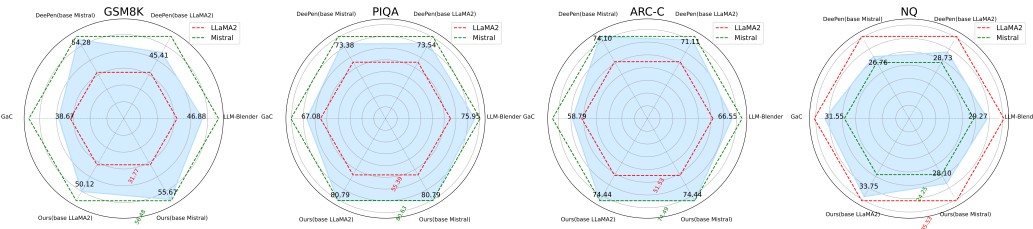

Figure 1: The impact of performance disparity among models on ensemble effectiveness across different datasets and methods is examined. We compare these methods to the individual performances of LLaMA2 and Mistral, indicated by dashed lines.

**Probability-Level Model Ensembling** methods focus on integrating outputs from different models by utilizing the probability distribution at each generation step. COOL-FUSION (Liu et al., 2024) let each source LLM generate tokens until reaching common word boundaries, then jointly reranks the segments. However, this method relies on common word boundaries, limiting generation flexibility and diversity. Additionally, managing multiple source models can increase complexity. DEEPEN (Huang et al., 2024), based on the principles of relative representation theory, transforms each model's probability distribution from its individual space into a common relative space for aggregation, which is determined through the intersection of the models' vocabularies. On the other hand, GAC (Yu et al., 2024b) constructs a mapping matrix that projects the probability vectors of multiple LLMs into the dimensional space of their union vocabulary, aggregating outputs to select the next token at each generation step. It is worth noting that all of these existing approaches operate on the entire vocabulary at every single step, resulting in significant computational overhead. In this work, we focus on the probability-level model ensembling without the need for additional training.

**Training-Level Model Ensembling** methods exemplified by FUSELLM (Wan et al., 2024a), leverage output probability vectors from various models during the training process, using these vectors as labels instead of one-hot representations. This technique serves as a unique form of distillation, enabling the trained model to extract richer information from the probability outputs of the ensemble of teacher models. FUSECHAT (Wan et al., 2024b) extends the framework of FUSELLM to fuse multiple chat LLMs with diverse architectures and scales. However, distillation is predominantly aimed at enhancing smaller models, which limits its effectiveness in further advancing larger models. On the other hand, EVA (Xu et al., 2024) addresses vocabulary discrepancies by learning token alignment across different vocabularies, utilizing overlapping tokens for assistance. However, vocabulary alignment should involve both output embedding alignment and weight alignment, EVA achieves vocabulary alignment solely by solving the mapping matrix of output embeddings.

## 3 UNDERSTANDING MODEL ENSEMBLING FROM MODEL CAPACITY, VOCABULARY SIZE AND TASK

Existing model ensembling approaches (Huang et al., 2024; Yu et al., 2024b) only focus on designing the model ensembling method, offering limited discussion on the selection of base models for ensembling. However, insights from prior research (Dong et al., 2023; Lee et al., 2023), indicate that not all model pairs are compatible to be combined, particularly when there are significant differences in size, vocabulary, and performance. In our preliminary experiments, we explored various factors that might affect the performance of model ensembling, including model size (e.g., 3B/7B/8B/13B/70B), model architecture (dense/sparse), performance discrepancies, tokenization strategies (BPE/WordPiece), vocabulary size (e.g., 102K/64K/32K) and tasks variations (e.g., text generation/QA/multiple choices). Finally, we identified three representative factors for further analyses, including performance discrepancy, vocabulary size, and tasks variations.

### 3.1 IMPACT OF MODEL PERFORMANCE DISCREPANCY

To investigate the impact of performance disparity on model ensembling, we select LLaMA2-13B-Chat and Mistral-7B-Instruct-v0.3 as base models, which show a significant performance gap across

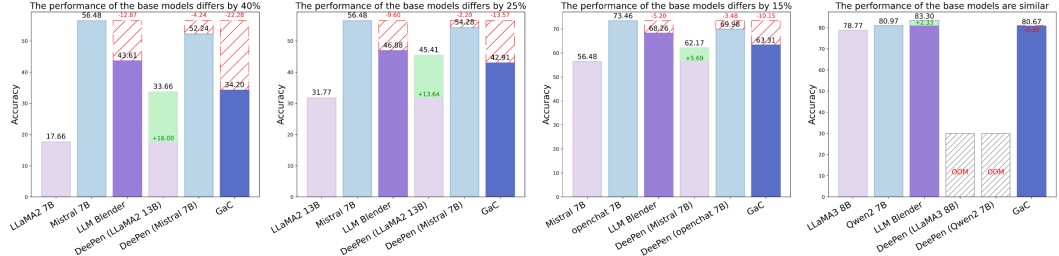

Figure 2: The impact of performance differences on model ensembling effectiveness on GSM8K dataset. OOM represents out of memory issue.

| Methods | Dataset | | Methods | Dataset | | Methods | Dataset | |
|---|---|---|---|---|---|---|---|---|
| | GSM | PIQA | | MMLU | ARC-C | | NQ | ARC-C |
| DeepSeek-7B(102K) | 59.67 | 72.66 | DeepSeek-7B(102K) | 46.97 | 58.73 | Mistral-7B(32K) | 24.25 | 74.49 |
| Mistral-7B(32K) | 56.48 | 80.63 | LLaMA2-13B(32K) | 49.61 | 51.53 | Yi-6B(64K) | 22.55 | 73.21 |
| LLM-BLENDER | 61.16 | 76.54 | LLM-BLENDER | 48.86 | 57.84 | LLM-BLENDER | 22.97 | 72.48 |
| | (+1.49) | (-4.09) | | (-0.75) | (-0.89) | | (-1.28) | (-2.01) |
| DEEPEN(DeepSeek-7B) | 55.00 | 77.65 | DEEPEN(DeepSeek-7B) | 52.81 | 60.00 | DEEPEN(Mistral-7B) | 25.26 | 76.50 |
| | (-4.67) | (+4.99) | | (+5.84) | (+1.27) | | (+1.01) | (+1.01) |
| DEEPEN(Mistral-7B) | 61.28 | 76.32 | DEEPEN(LLaMA2-13B) | 54.09 | 62.39 | DEEPEN(Yi-6B) | 22.80 | 77.86 |
| | (+4.80) | (-4.31) | | (+4.48) | (+10.86) | | (+0.25) | (+4.65) |
| Ours(DeepSeek-7B) | 62.77 | 81.81 | Ours(DeepSeek-7B) | 48.90 | 60.16 | Ours(Mistral-7B) | 24.76 | 76.54 |
| | (+3.10) | (+9.15) | | (+1.93) | (+1.43) | | (+0.51) | (+2.05) |
| Ours(Mistral-7B) | 58.88 | 81.81 | Ours(Mistral-7B) | 48.90 | 60.16 | Ours(Yi-6B) | 23.28 | 76.54 |
| | (+2.40) | (+1.18) | | (-0.71) | (+8.63) | | (+0.73) | (+3.33) |

Table 1: The influence of vocabulary size on model ensembling effectiveness

various tasks. We evaluated the GSM8K (Cobbe et al., 2021), PIQA (Bisk et al., 2020), ARC-C (Clark et al., 2018), and NQ (Kwiatkowski et al., 2019) datasets with three comparative methods.

As shown in Fig. 1, we found that when there is a significant performance disparity between models, LLM-BLENDER, which selects the optimal response from model candidates, is not suitable for model ensembling. Similarly, for methods like DEEPEN and GAC, which are achieved through token probability averaging, a substantial performance gap also leads to certain performance drops. This observation is straightforward and easily understood, as lower-performing models may introduce noise and bias, adversely affecting overall effectiveness. Nonetheless, it has been observed that DEEPEN and GAC consistently enhance the performance of less effective models when combined with a superior model. Although the resultant performance does not surpass that of the superior model, these methods can also serve as an alternative way for training-free knowledge distillation.

To further figure out how performance discrepancy impacts the ensembling, we then identify model pairs based on the GSM8K dataset with performance gaps of approximately 40% (LLaMA2-7B-Chat and Mistral-7B-Instruct-v0.3), 25% (LLaMA2-13b-Chat and Mistral-7B-Instruct-v0.3), 15% (OpenChat-3.5 and Mistral-7B-Instruct-v0.3), and less than 10% (LLaMA3-8B-Instruct and Qwen2-7B-Instruct). Fig. 2 shows that as the performance gap increases, the improvement from ensembling inferior models becomes more pronounced. In contrast, for superior models, the ensembling effect consistently falls below the baseline performance. When the performance difference is within 10%, ensembling may yield better results.

Moreover, it was observed that with the increase in vocabulary size, as exemplified by models such as LLaMA3 and Qwen2, whose vocabulary size exceeds 120K, approaches like DEEPEN, which depend on high-dimensional matrix operations, encounter heavy memory limitations that diminish their effectiveness.

**TAKEAWAY I: Smaller performance gaps lead to greater gains from model ensembling.**

## 3.2 INFLUENCE OF VOCABULARY SIZE

Yu et al. (2024b) found that large models employing different tokenization methods, such as BPE and BBPE, exhibit over 90% overlap in tokenizing Oxford 5000 common words. This indicates that tokenization methods have a marginal impact on model ensembling, leading us to focus on the effect

| Question | who wrote he ain't heavy he's my brother lyrics
who was the ruler of england in 1616 |
|---|---|
| Gold answer | Bob Russell
James I |
| LLaMA3 response | Bob Russell ✔
James I ✔ |
| Qwen2 response | He Ain't Heavy, He's My Brother was written by Mike D'Abo. ✗
The song was originally recorded by the British singer-songwriter and released as a single in 1969.
It became a hit and has since been covered by numerous artists.
. . .
---
The ruler of England in 1616 was King James I. ✔
He was the first monarch of the Stuart dynasty and reigned from March 24, 1603, until his death on July 19, 1625.
. . . |
| LLM-Blender response | He Ain't Heavy, He's My Brother was written by Mike D'Abo. ✗
The song was originally recorded by the British singer-songwriter and released as a single in 1969.
It became a hit and has since been covered by numerous artists.
. . .
---
The ruler of England in 1616 was King James I. ✔
He was the first monarch of the Stuart dynasty and reigned from March 24, 1603, until his death on July 19, 1625.
. . . |

Table 2: Responses of different methods on NQ dataset.

of vocabulary size on ensemble performance. To this end, we selected four models with varying vocabulary sizes that exhibit similar performance on certain datasets, namely LLaMA2-13B-Chat (vocabulary size: 32,000), Mistral-7B-Instruct-v0.3 (vocabulary size: 32,768), Yi-6B (vocabulary size: 64,000), and DeepSeek-LLM-7B-Chat (vocabulary size: 102,400).

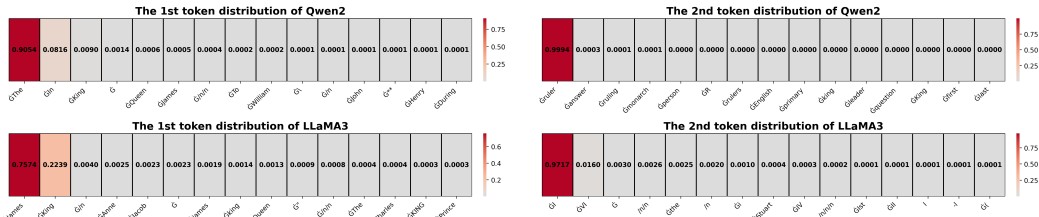

Figure 3: Token distribution of different models. We extract the top 15 words from the vocabulary of each model and apply a softmax processing to their corresponding logits.

As presented in Table 1, existing methods demonstrate consistent performance irrespective of the gap in vocabulary size, thereby indicating that vocabulary size does not significantly influence the efficacy of model ensembling. To elucidate the underlying causes, we conducted a manual analysis of 100 responses generated by various LLMs. Our investigation revealed that approximately 80% of the words in the responses were common words while the tokenization of these common words exhibited no discernible differences across the different LLMs. Therefore, the vocabulary size, as well as the tokenization strategy, exerts minimal influence on the performance of model ensembling. Notably, despite significant variations in vocabulary size, the ensembling results of models with similar performance suggest that when the performance gap between base models is within 10%, model ensembling may achieve better results than the baseline model, which is aligned with findings in Section 3.1. More interestingly, results of evaluating DEEPPEN on the ARC-C dataset indicate that when model performances are comparable, ensembling inferior models can even outperform superior ones. This might imply that when performance differences are minimal, the selection of main base models does not necessarily need to prioritize those with superior performance. Besides, it is important to note that these findings related to the vocabulary and tokenization consistently apply to LLM-BLENDER, DEEPEN and our method, with the exception of GAC. We will detail the reason in Section 4.2.

**TAKEAWAY II: The influence of vocabulary size for model ensembling is marginal.**

### 3.3 TASK-SPECIFIC CHALLENGES IN MODEL ENSEMBLING

In the aforementioned experiments, we identified an interesting phenomenon: Even when performance and vocabulary size are aligned across models, substantial differences in the response style can also hinder successful ensembling on specific tasks. For example, LLaMA3-8B and Qwen2-7B exhibit comparable performance across various tasks. Existing ensembling approaches such as LLM-BLENDER have been shown to enhance outcomes on GSM8K and PIQA tasks when utilizing these two base models. However, ensembling these models to address the questions in TriviaQA and NQ datasets proves to be impractical due to the significant discrepancies in their response styles.

As shown in Table 2, under identical prompting conditions, LLaMA3 tends to directly return the answers, whereas Qwen2 is inclined to conduct analysis. In voting-based LLM-BLENDER, response length significantly influences judgments, with PAIRRANKER consistently favoring longer responses. For instance, in our analysis of 100 randomly selected NQ and TriviaQA responses, the PAIRRANKER favored the longer response as the answer in approximately 80% and 70% of cases accordingly, possibly due to its training data characteristics. More detailed analysis of TriviaQA and more experimental results are in appendix A. Similarly, probability-based ensembling methods like DEEPEN and GAC are also affected by the differences in the reasoning process, leading to significant deviations in vocabulary probability distributions. Figure 3 reveals a significant disparity in top token distributions between LLaMA3 and Qwen2 for the first two tokens. Averaging the probabilities directly could result in an excessively smoothed distribution, which is inappropriate. The underlying reason for this phenomenon may be the differences in LLMs' training data. Therefore, caution is warranted in specific tasks to avoid substantial disparities, despite effective ensembling in most other tasks.

**TAKEAWAY III: Even though the performance and vocabulary size are aligned across models, substantial differences in response style could also hinder successful ensembling.**

## 4 METHODOLOGY

### 4.1 MODEL SELECTION STRATEGY

Given the aforementioned insights, we present a strategy to assess the compatibility of two models for ensembling and offer guidance for selecting base models from the candidate model pool.

When ensembling two models, we begin by comparing their performance on the target tasks. An overall performance gap within 10% is desirable, as it indicates the potential compatibility for successful ensembling. If the performance gap falls within this range, it is crucial to compare the response styles of different models. This comparison is relatively subjective, with differences often identifiable through superficial features such as text length and BLEU score.

In this work, we use text length as a guiding metric by constraining the length of longer responses to not exceed twice the length of the shorter ones. When the response manner is also consistent across models, it is highly probable that these two models can be successfully ensembled to produce a superior response. We exclude discrepancies in vocabulary size from consideration, as prior analyses have demonstrated its insignificance in affecting ensembling outcomes.

To select base models from the pool, we recommend initially choosing the best-performing model for the target tasks. Subsequently, select the next best-performing model that satisfies the criteria for successful ensembling with the first chosen model, continuing this process iteratively until the maximum number of base models is reached or no further suitable models can be found.

### 4.2 UNION TOP-$k$ ENSEMBLING

Apart from the model selection strategy, we also endeavor to improve the efficiency and performance of model ensembling. Existing probability-level model ensembling methods, e.g. DEEPEN and GAC, attempt to combine the models through full vocabulary alignment. We argue this approach is sub-optimal since the candidate next token typically resides within the top-k tokens. Conversely, incorporating tokens of lower probability may introduce unnecessary noise, thereby diminishing the overall accuracy and effectiveness of the ensemble. Motivated by this, we propose the UNION Top-$k$

Ensembling (UNITE), a novel and efficient ensembling approach that only aligns the top-$k$ tokens in each decoding step.

Given the base models for ensembling, denoted as LLM$_i$ with their respective tokenizer $T^i$ and vocabulary $V^i$, we feed the input text *prompt* into LLM$_i$ and obtain the probability distribution. Instead of aligning the probability distributions across different models, we only focus on the subset of top-$k$ tokens since the candidate next token is highly likely to be within the subset. The selected top-$k$ tokens of LLM$_i$ is presented as TopK$_i$ = $(t_1^i, t_2^i, ..., t_k^i)$ with corresponding probability distribution $P^i = (p_1^i, p_2^i, ..., p_k^i)$.

Next, we aim to construct the union set $V^u$ of TopK$_i$ across the base models. The most straightforward solution to obtain the union set is to directly remove the duplicate tokens and retain all other distinct tokens. For the token in $V^u$ but not in $V^i$, the probability of this token in LLM$_i$ is assigned to 0. This strategy is adopted by GAC (Yu et al., 2024b). However, we contend that this strategy is not entirely reasonable since it is heavily influenced by the tokenizer and vocabulary. For example, the candidate token "James" is within the top-$k$ subset of LLM$_1$ but not in $V^2$ while the token "Jam" appears in TopK$_2$. The word "James" can be tokenized into "Jam" and "es" by LLM$_2$. If the token probabilities $p_{\{Jam\}}^1$ and $p_{\{James\}}^2$ are set to 0, the overall probability of generating the word "James" should be inherently reduced, even if both base models exhibit a preference for this word. Therefore, we present a new token probability alignment strategy on the union set. Firstly, we obtain the top-$k$ union vocabulary $V^u$ by directly merging the TopK$_i$ subsets. Then, the token distribution $P^i$ and TopK$_i$ are updated according to following criteria:

1. If the token $w$ appears in both $V^u$ and TopK$_i$, $P_{\{w\}}^i$ and TopK$_i$ remains unchanged.

2. If the token $w$ in $V^u$ is absent in TopK$_i$ but present in $V^i$, it will be appended to TopK$_i$. $P^i$ also updates accordingly.

3. If the token $w$ in $V^u$ does not exist in $V^i$, it should be tokenized by $T^i$. The first token of the result along with its token probability is then updated to TopK$_i$ and $P^i$.

Up to now, we can obtain the aligned top tokens Top$\hat{K}_i$ and the token distributions $\hat{P}^i$ of based models. Consequently, we normalize $\hat{P}^i$ as: $\hat{P}_{norm}^i = \text{softmax}(\hat{P}^i)$.

Since our method eliminates the need for full vocabulary alignment, it is essential to designate one model as the primary base model. As discussed in Section 3.2, the selection of primary base model is flexible when the candidate models exhibit comparable performance. To simplify the process, we directly employ the best-performing model as the primary base model. Then, the primary base model employs the average token probability $\hat{P}_{avg}$ to predict the next token, where $\hat{P}_{avg} = \frac{1}{n} \sum_{i=1}^{n} \hat{P}_{norm}^i$.

The next token is determined using the maximization-based greedy strategy (Li et al., 2016). The chosen token will be appended to the input text. This process is iteratively repeated until the predetermined stopping criterion is met, such as generating an end-of-sentence token or reaching the maximum length. The algorithm process can be referenced in the Appendix, Algorithm 1.

## 5 EXPERIMENTS

### 5.1 SETUP

**Models**   All of our experiments are conducted with the following commonly used models, including LLaMA2-7B-Chat (Touvron et al., 2023), LLaMA2-13B-Chat (Touvron et al., 2023), LLaMA3-8B-Instruct (Dubey et al., 2024), LLaMA3.1-8B-Instruct (Dubey et al., 2024), Mistral-7B-Instruct-v0.3 (Jiang et al., 2023a), DeepSeek-LLM-7B-Chat (Bi et al., 2024), Yi-6B (Young et al., 2024), OpenChat-3.5 (Wang et al., 2024a), Qwen2-7B-Instruct (Yang et al., 2024), Mixtral-8×7B (Jiang et al., 2024) and Qwen1.5-72B (Bai et al., 2023).

**Baselines**   We selected three typical model ensembling methods for further analyses. 1) **LLM-BLENDER** (Jiang et al., 2023b) includes a reward model, PAIRRANKER, to evaluate LLM responses, and a fusion model, GENFUSER, to combine them. We only use PAIRRANKER due to significant over-generation issues with GENFUSER. 2) **DEEPEN** (Huang et al., 2024) employs relative representation theory to map each model's probability distribution to a universal space for aggregation,

| Method | Dataset | | | | | | Avg. |
|---|---|---|---|---|---|---|---|
| | GSM8K | PIQA | MMLU | ARC-C | TriviaQA | NQ | |
| Mistral | 56.48 | 80.63 | 59.28 | 74.49 | 64.30 | 24.25 | 59.91 |
| DeepSeek | 59.67 | 72.66 | 46.97 | 58.73 | 47.63 | 11.37 | 49.51 |
| OpenChat | 73.46 | 87.10 | 60.80 | 78.05 | 61.77 | 31.08 | 65.38 |
| LLM-BLENDER | 70.79(-2.67) | 83.28(-3.82) | 60.10(-0.70) | 76.29(-1.76) | 56.35(-5.42) | 25.57(-5.51) | 62.06(-3.32) |
| DEEPEN | 73.06(-0.40) | 76.04(-11.06) | 61.91(+1.11) | 72.14(-5.91) | 67.24 (+5.47) | 28.26(-2.82) | 63.11(-2.27) |
| GAC | 62.85(-10.61) | 67.85(-19.25) | 55.15(-5.65) | 73.04(-5.01) | 62.22(+0.45) | 20.72(-10.36) | 56.97(-8.41) |
| UNITE | 73.31(-0.15) | 87.50 (+0.40) | 62.13 (+1.33) | 78.70 (+0.65) | 65.80(+4.03) | 31.78 (+0.70) | 66.54 (+1.16) |

Table 3: Results of ensembling on Mistral, DeepSeek, OpenChat. OpenChat is chosen as primary model for these experiments.

| Method | Dataset | | | | Avg. |
|---|---|---|---|---|---|
| | GSM8K | PIQA | ARC-C | MMLU | |
| LLaMA3 | 78.77 | 79.08 | 79.01 | 64.58 | 75.36 |
| LLaMA3.1 | 80.83 | 82.86 | 79.49 | 66.69 | 77.47 |
| Qwen2 | 80.78 | 84.57 | 84.92 | 64.96 | 78.81 |
| *Two-model ensembling (LLaMA3+Qwen2)* | | | | | |
| LLM-BLENDER | 82.69 (+1.91) | 82.53 (-2.04) | 82.98 (-1.94) | 62.07 (-2.89) | 77.57 (-1.24) |
| DEEPEN | - OOM - | | | | |
| GAC | 80.67 (-0.11) | 80.96 (-3.61) | 84.93 (+0.01) | 67.05 (+2.09) | 78.40 (-0.41) |
| UNITE | 84.17 (+3.39) | 85.53 (+0.96) | 85.07 (+0.15) | 69.78 (+4.82) | 81.14 (+2.33) |
| *Three-model ensembling* | | | | | |
| LLM-BLENDER | 83.30(+2.52) | 83.47(-1.10) | 83.48(-1.44) | 62.55(-2.41) | 78.20(-0.61) |
| DEEPEN | - OOM - | | | | |
| UNITE | 84.99 (+4.21) | 84.98 (+0.41) | 85.39 (+0.47) | 69.12 (+4.16) | 81.12 (+2.31) |

Table 4: Results of ensembling on LLaMA3, LLaMA3.1, Qwen2. Qwen2 is chosen as primary model for these experiments.

computed through the intersection of model vocabularies. 3) **GAC** (Yu et al., 2024b) projects multiple LLMs' probability vectors into a unified vocabulary dimension using a mapping matrix and aggregates outputs at each generation step to select the next token. Although GAC suggests the importance of identifying keywords to improve latency, we still exclude it owing to its hindrance on the ensembling performance.

**Benchmarks**  We evaluate six benchmarks, which can be categorized into three main groups. 1) **Comprehensive Examination:** MMLU (5-shot) (Hendrycks et al., 2021), covering 57 subjects that humans typically learn; ARC-C (0-shot) (Clark et al., 2018), collected from standardized natural science tests. 2) **Reasoning Capabilities:** GSM8K (Cobbe et al., 2021) (4-shot), a dataset of high-quality problems at the grade school math level; PIQA (Bisk et al., 2020) (0-shot), a commonsense reasoning dataset. 3) **Knowledge Capacities:** TriviaQA (5-shot) (Joshi et al., 2017), compiled by Trivia enthusiasts; NaturalQuestion (NQ) (5-shot) (Kwiatkowski et al., 2019), a question-answering corpus consisting of queries issued to the Google search engine.

For a comprehensive evaluation, we conduct two- and multi-model ensembling experiments on the aforementioned benchmarks. We select base models following the strategy presented in Section 4.1. We choose two pairs from Mistral, Deepseek, and OpenChat for the two-model experiments. For the closely performing LLaMA3, LLaMA3.1, and Qwen2, we select LLaMA3 and Qwen2 for the two-model ensemble based on their release dates and subsequently employ all three models for the multi-model ensembling evaluations. The hyper-parameter $k$ is set to 10 in this work. All experiments are conducted on 46G NVIDIA L40 GPUs.

## 5.2 MAIN RESULTS

As shown in Table 3 and Table 4, we have following observations:

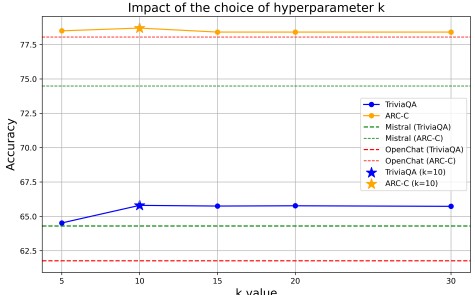
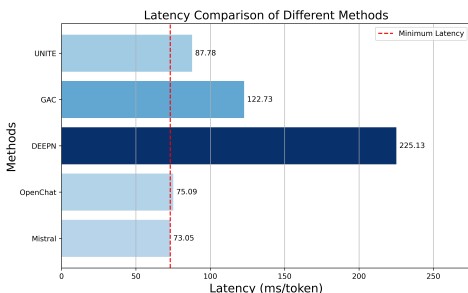

Figure 4: Impact of the choice of hyperparameter $k$ on the ARC and TriviaQA datasets. Increasing $k$ beyond a certain point leads to a slight decline or no improvement in performance.

Figure 5: Latency comparison of different methods. The darker the color, the greater the latency.

**(1) UNITE enhances individual model performance when the base models exhibit similar performance levels.** As demonstrated, the ensemble of models such as OpenChat achieved an average improvement of approximately 1.2% across five benchmark tasks, including ARC-C and MMLU. However, in the GSM8K task, a 15% performance gap between DeepSeek and OpenChat led to a slight decline in overall performance, reinforcing our observation in Section 3.1 that ensemble methods are most effective when model performances are closely aligned. Furthermore, experiments with LLaMA3 and Qwen2 validated our approach, resulting in performance increases of 3.39% and 4.82% on the GSM8K and MMLU tasks, respectively.

**(2) UNITE demonstrates greater robustness and generality.** Ensembling experiments with LLaMA3 and Qwen2 indicate that while LLM-BLENDER improves performance on the GSM8K dataset, it significantly underperforms compared to baseline models in the PIQA, ARC-C, and MMLU benchmarks. Specifically, BLENDER demonstrates a decline of 2.89% in the MMLU task, and this issue is exacerbated in ensembles with the baseline models OpenChat and Mistral, which exhibit relatively low baseline performance, leading to an average performance drop of 3.32%. Additionally, GAC fails to improve the performance of individual models across most tasks and exhibits a significant decline of over 10% when combined with Mistral and OpenChat. Furthermore, due to its intersection limitations, DEEPEN is ill-suited for ensembling with the LLaMA3 model, which requires handling a large vocabulary. In contrast, across tasks with comparable performance levels, UNITE achieves the highest improvements and overall performance across the board, with the exception of a slight under-performance relative to DEEPEN on TriviaQA. This outcome underscores the effectiveness and robustness of UNITE.

**(3) Collaborating with comparable LLMs does not necessarily yield better results.** The ensembling experiments conducted with three models, following the integration of LLaMA3.1, demonstrate improved performance on the GSM8K and ARC-C benchmarks compared to the ensemble of LLaMA3 and Qwen2. However, this approach results in suboptimal outcomes on PIQA and MMLU. This observation is justified, as the analysis in Section 3.2 indicates that while combining models with similar performance may enhance overall efficacy, such improvement is not guaranteed.

## 5.3 ABLATION STUDY

**Effect of hyperparameter $k$ selection**    To investigate the impact of the hyperparameter $k$, we conducted experiments using the Mistral and OpenChat models on the TriviaQA and ARC-C datasets. As illustrated in Fig. 4, increasing $k$ from 5 to 10 enhances ensemble performance. However, further increasing $k$ beyond 10 leads to either a slight decline or no change in performance. This finding supports our assertion that, in probability-level ensembling, aligning the entire vocabulary is unnecessary for predicting the next token.

**Effect of the next token selection strategy**    We also explored the impact of deterministic decoding and top-$k$ sampling (Fan et al., 2018) on the next token emission. It is important to clarify that our UNITE focuses on the fact that, when predicting the next token, we only need to ensemble a subset of tokens rather than the entire vocabulary. In contrast, greedy decoding and top-$k$ sampling emphasize how to determine the next token's ID after the ensemble process is complete. To investigate

| Base Model | Method | Tokens Manipulated Each Step |
|---|---|---|
| Mistral(32678) + OpenChat(32002) | DEEPEN | 32000 |
| | GAC | 32770 |
| | UNITE | 14.46 |
| LLaMA3(128256) + Qwen2(151646) | DEEPEN | 109566 |
| | GAC | 170336 |
| | UNITE | 14.43 |

Table 5: Tokens manipulated at each step. It's noteworthy that DEEPEN encountered out-of-memory issues when conducting LLaMA3 and Qwen2 ensembling.

| Model | ARC-C | PIQA |
|---|---|---|
| Qwen1.5-72B(Dense) | 69.03 | 73.83 |
| Mixtral-8x7B(Sparse) | 73.98 | 74.59 |
| UNITE | 78.84 | 81.88 |
| | (+4.86) | (+7.29) |

Table 6: Ensemble learning of the dense LLM Qwen1.5-72B and the sparse MOE model Mixtral-8x7B.

this, we conducted experiments using LLaMA3.1 and Mistral on the PIQA and ARC-C datasets correspondingly. As shown in Figure 6, for these deterministic tasks, the maximization-based greedy method outperforms the random sampling approach, which aligns with intuition.

## 5.4 FURTHER ANALYSIS

**Latency analysis** Following the settings detailed in GAC (Yu et al., 2024b), we recorded the latency (*ms/token*) of different methods. The data presented in Fig. 4 reveals significant differences in latency among the various methods. Notably, the latency of UNITE is 87.78 *ms/token*, which is substantially lower than that of DEEPEN and GAC, and only about 10 *ms* longer than that of the individual models.

**Tokens manipulated each step** Table 5 presents tokens manipulated at each step. Compared to DEEPEN and GAC, UNITE significantly reduces the number of tokens manipulated at each step, indicating its efficiency. As the vocabulary of the base model expands, the number of manipulated tokens of DEEPEN and GAC increases significantly, which not only increases the computational burden but may also lead to performance bottlenecks. Contrastly, UNITE is minimally affected, maintaining its effectiveness.

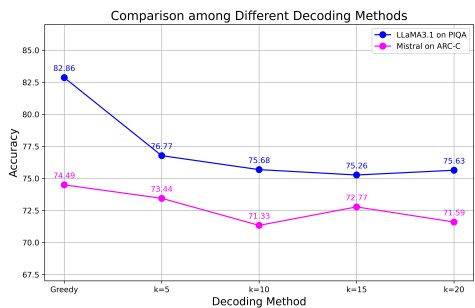

Figure 6: Comparison among different decoding methods. The greedy decoding strategy is more effective for eliciting the next token in deterministic tasks.

**Ensemble of the dense model and the sparse model** We evaluate our method on the ensemble learning of the dense model and the sparse MoE model for reasoning and comprehensive examination tasks. Specifically, we utilize the widely-used large-scale dense model Qwen1.5-72B (Bai et al., 2023) alongside the popular sparse MoE model Mixtral-8×7B (Jiang et al., 2024) as our base models. As shown in Tabel 6, our UNITE achieves +4.86% and +7.29% performance on the ARC-C and PIQA datasets respectively, despite the base models already exhibiting high levels of performance.

## 6 CONCLUSION

In conclusion, our research highlights the effectiveness of ensemble methods in enhancing the performance of LLMs. By examining existing techniques, we identified key factors influencing ensembling success, such as model performance and response processes, while finding that vocabulary size has a minimal impact. In addition, the issue of vocabulary redundancy exposed by existing methods during ensembling lead us to propose UNITE, which efficiently aggregates a bunch of tokens from multiple LLMs without the computational overhead. Through extensive experimentation, UNITE consistently outperformed state-of-the-art ensemble methods, demonstrating its effectiveness in leveraging the strengths of diverse LLMs. Our contributions not only advance the understanding of model ensembling but also provide a practical framework for selecting and integrating LLMs to achieve superior performance.

ACKONWLEDGEMENTS

This work was supported in part by the Research Grants Council of the Hong Kong SAR under Grant GRF 11217823 and Collaborative Research Fund C1042-23GF, the National Natural Science Foundation of China under Grant 62371411, InnoHK initiative, the Government of the HKSAR, Laboratory for AI-Powered Financial Technologies.

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

# A APPENDIX

## A.1 ALGORITHM OF UNITE

As presented in Algorithm 1, UNITE integrates the Top-k tokens generated by multiple base models to create a joint set, updating the tokens based on probability rules, and ultimately uses a greedy strategy to select the next token from this set, repeating the process until a stopping criterion is met.

---

**Algorithm 1** Union top-$k$ ensembling

---

**Require:** $\text{LLM}_i$, Vocabulary $V_i$, Tokenizer $T_i$, Demonstration $prompt$, Stop condition $stop(*)$
1: **while** not $stop(*)$ **do**                                                      ▷ Stopping criteria
2:     $\text{TopK}_i, P^i \leftarrow \text{LLM}_i(prompt)$                 ▷ Generate top-$k$ tokens and their probabilities
3:     **for** each model **do**
4:       **if** token $w \in V^u$ and $w \in \text{TopK}_i$ **then**
5:         $P^i_{\{w\}}$ and $\text{TopK}_i$ remains unchanged.
6:       **else if** token $w \in V^u$ and $w \in V^i$ and $w \notin \text{TopK}_i$ **then**
7:         $\hat{\text{TopK}}_i \leftarrow w, \hat{P}^i \leftarrow p^i_w$
8:       **else if** token $w \in V^u$ and $w \notin V^i$ **then**
9:         $w^1, \ldots, w^m \leftarrow T^i(w)$
10:        $\hat{\text{TopK}}_i \leftarrow w^1, \hat{P}^i \leftarrow p^i_{w^1}$
11:       **end if**
12:     **end for**
13:     $\hat{P}^i_{norm} = \text{softmax}(\hat{P}^i), \hat{P}_{avg} = \frac{1}{n} \sum_{i=1}^{n} \hat{P}^i_{norm}$
14:     $w = \underset{w \in Top\hat{K}_i}{\arg\max} (\hat{P}_{avg})$                    ▷ Predict the next token
15:     $prompt \leftarrow prompt + w$                                        ▷ Update input sequence
16: **end while**

---

## A.2 TASK CHALLENGES ANALYSIS ON TRIVIAQA

Table 7 provides a detailed comparison of the differences in response styles between LLaMA3 and Qwen2 on the TriviaQA dataset. It is easily observable that, regardless of correctness, LLM-BLENDER consistently tends to choose longer responses as answers.

We try to address the response style issue via preprocessing steps. Specifically, we provide an alternative simple solution by using the few-shot examples to standardize the response format. We employed a new 5-shot prompt designed to elicit answers in the format: The answer is xxx. The responses are presented in the Table 8.

As shown in Table 9, the original prompts elicit accuracy in the brackets. Since the tedious response style illustrated in Table 7, Qwen incorporates answers into the analysis, we randomly sampled 100 instances from the 1500-test set to manually extract the predictions. After adjusting the prompt, we can easily analyze the results for the entire test set. Consistent with our findings presented in the main text, when the base model's performance gap exceeds 10%, ensemble learning may yield little to no improvement. Additionally, it is important to note that our UNITE approach still outperforms its competitors.

## A.3 GENERALIZATION EVALUATION ON BIG-BENCH HARD BENCHMARK

We conduct additional experiments using the BBH (BIG-Bench Hard) benchmark (Suzgun et al., 2022), a diverse evaluation suite of 23 challenging tasks such as symbolic reasoning, to further validate UNITE's generalizability. Due to computational constraints and limited rebuttal time, we randomly selected 10 subsets, and the results are presented below. Consistent with our main findings, UNITE demonstrates superior performance compared to other methods, highlighting the effectiveness and generalizability of our approach.

| Question | Who had a 70s No 1 hit with Kiss You All Over? |
| | The flag of Libya is a plain rectangle of which color? |
| Gold answer | Sir Henry Campbell Bannerman |
| | Green |
| LLaMA3 response | Henry Campbell-Bannerman ✔ |
| | Green ✔ |
| Qwen2 Response | The hit song "Kiss You All Over" was performed by the band Arrows, featuring vocalist Jim McCarty. ✗ |
| | The song reached number one on the UK Singles Chart in 1976. |
| | The song was written by Arrows' lead singer Jim McCarty, along with the band's guitarist, Steve "Pig" Harrison. |
| | It was produced by Tony Visconti, who had previously worked with David Bowie and T. Rex. |
| | "Kiss You All Over" was released as a single from the band's debut album, "Arrows," which was also released in 1976. |
| | The song's catchy chorus and upbeat tempo made it a popular hit of the 1970s, |
| | and it remains a well-known and beloved track from that era. |
| | The song's success helped to establish Arrows as a popular band of the time, |
| | and it continues to be played on classic rock and oldies radio stations today. |
| | The flag of Libya is a plain rectangle of green color. ✔ |
| | It was adopted in 1977 and features a green color field with no other symbols or designs. |
| | The color green is significant in the country's history and culture, |
| | representing hope, fertility, and the country's rich agricultural heritage. |
| | The flag is rectangular in shape and is often flown vertically. |
| | It is an important symbol of the Libyan state and is used in various official and ceremonial contexts. |
| LLM-Blender response | The hit song "Kiss You All Over" was performed by the band Arrows, featuring vocalist Jim McCarty. ✗ |
| | The song reached number one on the UK Singles Chart in 1976. |
| | The song was written by Arrows' lead singer Jim McCarty, along with the band's guitarist, Steve "Pig" Harrison. |
| | It was produced by Tony Visconti, who had previously worked with David Bowie and T. Rex. |
| | "Kiss You All Over" was released as a single from the band's debut album, "Arrows," which was also released in 1976. |
| | The song's catchy chorus and upbeat tempo made it a popular hit of the 1970s, |
| | and it remains a well-known and beloved track from that era. |
| | The song's success helped to establish Arrows as a popular band of the time, |
| | and it continues to be played on classic rock and oldies radio stations today. |
| | The flag of Libya is a plain rectangle of green color. ✔ |
| | It was adopted in 1977 and features a green color field with no other symbols or designs. |
| | The color green is significant in the country's history and culture, |
| | representing hope, fertility, and the country's rich agricultural heritage. |
| | The flag is rectangular in shape and is often flown vertically. |
| | It is an important symbol of the Libyan state and is used in various official and ceremonial contexts. |

Table 7: Responses of different methods on TriviaQA dataset

| TriviaQA Question | Which Lloyd Webber musical premiered in the US on 10th December 1993? |
| Original prompt (Response style referring to Table 2) | Question: In the 1971 Number One hit Ernie by Benny Hill, |
| | what was the name of Ernie's horse who was kicked by his rival, |
| | Two-ton Ted from Teddington? |
| | Answer: **Triggers.** |
| | . . . |
| New Prompt | Question: In the 1971 Number One hit Ernie by Benny Hill, |
| | what was the name of Ernie's horse who was kicked by his rival, |
| | Two-ton Ted from Teddington? |
| | Answer: **The answer is Triggers.** |
| | . . . |
| LLaMA3 response | The answer is Sunset Boulevard. |
| Qwen2.5 response | The answer is Sunset Boulevard. |

Table 8: Case study of preprocessing on TriviaQA.

## A.4 Multilingual Tokenization Considerations

We also conducted experiments on the CMMLU (Li et al., 2024), a Chinese Multitask Language Understanding Evaluation benchmark to validate our findings in section 3.2. We used Yi-6B (vocabulary size: 64,000) and Qwen2-7b-instruct (vocabulary size: 152,064) as our base models, both supporting the Chinese language. Qwen2-7b-instruct is the primary model, the results of various ensemble approaches are presented in Table 11. Similar to the results listed in Section 3.2, irrespective

| Method | TriviaQA |
|---|---|
| LLaMA3-8b-instruct | 70.68(67) |
| Qwen2.5-7b-instruct | 57.85(52) |
| LLM-BLENDER | 64.77 |
| UNITE | 67.45 |

Table 9: Results of ensembling LLaMA3 and Qwen2.5 on TriviaQA after preprocessing. LLaMA3 is chosen as primary model for these experiments.

| Method | BBH |
|---|---|
| LLaMA3-8b-instruct | 73.00 |
| Qwen2-7b-instruct | 68.60 |
| LLM-BLENDER | 68.79 |
| DEEPEN | -OOM- |
| GAC | 69.86 |
| UNITE | 73.52 |

Table 10: Results of ensembling LLaMA3 and Qwen2 on BBH. LLaMA3 is chosen as primary model for these experiments.

| Method | CMMLU |
|---|---|
| Yi-6B | 75.21 |
| Qwen2-7b-instruct | 83.22 |
| LLM-BLENDER | 79.08 |
| DEEPEN | -OOM- |
| GAC | 75.88 |
| UNITE | 83.89 |

Table 11: Results of ensembling Yi and Qwen2 on CMMLU. Qwen2 is chosen as primary model for these experiments.

of the gap in vocabulary size, existing methods still demonstrate improvements, thereby indicating that vocabulary size for model ensembling is marginal.

## A.5 FURTHER ANALYSIS OF HYPERPARAMETER $k$

We clarify that the enhanced efficiency and effectiveness arise from reduced token options and our specialized union mapping method, as outlined in Section 4.2. UNITE constructs a union of the top-$k$ tokens from each model and expands this set using each model's tokenizer, followed by probability aggregation to determine the next token. UNITE eliminates the need for auxiliary mapping matrices and full vocabulary alignment, respecting the unique tokenization of each base LLM.

We further test $k$ with extremely large values to validate our effectiveness. As shown in Table 12, further increasing $k$ leads to either a slight decline or no change in performance. This finding reinforces our assertion that, in probability-level ensembling, it is unnecessary to align the entire vocabulary to predict the next token.

## A.6 CASE STUDY OF UNITE

Table 13 presents the specific cases of UNITE and other ensembling approaches, showcasing the validity and stability of our method.

| Setting | TriviaQA |
|---|---|
| Mistral | 64.30 |
| OpenChat | 61.77 |
| $k = 5$ | 64.52 |
| $k = 10$ | 65.80 |
| $k = 20$ | 65.77 |
| $k = 100$ | 65.65 |
| $k = 1000$ | 65.74 |
| $k = 10000$ | 65.72 |

Table 12: Impact of the large hyperparameter $k$ on TriviaQA dataset.

| | | |
|---|---|---|
| Question | | Tom plants 10 trees a year. Every year he also chops down 2 trees a year. He starts with 50 trees. After 10 years 30% of the trees die. How many trees does he have left? |
| Gold Answer | | He gets $10 - 2 = << 10 - 2 = 8 >>$ 8 new trees a year. After 10 years he has $8 * 10 = << 8 * 10 = 80 >>$ 80 new trees so he had $80 + 50 = << 80 + 50 = 130 >>$ 130 trees He lost $130 * .3 = << 130 * .3 = 39 >>$ 39 trees That means he had $130 - 39 = << 130 - 39 = 91 >>$ `91` trees |
| LLaMA3 Response | | Tom starts with 50 trees. He plants 10 trees a year, so after 10 years he has 50+10*10= 150 trees. However, he chops down 2 trees a year, so after 10 years he has 150-2*10= 140 trees. 30% of the trees die, which means 30/100*140= 42 trees die. Tom is left with 140-42= 98 trees. The answer is boxed{ `98` } |
| LLaMA3.1 Response | | Tom starts with 50 trees and plants 10 trees a year, so after 10 years he will have 50+10*10= 150 trees. However, he also chops down 2 trees a year, so after 10 years he will have 150-2*10= 140 trees. After 10 years, 30% of the trees die, so 30% of 140 is 0.3*140= 42 trees die. The number of trees Tom has left is 140-42= 98 trees. The answer is boxed{ `98` } |
| Qwen2 Response | | Tom starts with 50 trees and plants 10 trees a year, so after 10 years he will have 50+10*10= 150 trees. However, he also chops down 2 trees a year, so after 10 years he will have 150-2*10= 140 trees. After 10 years, 30% of the trees die, so 30% of 140 is 0.3*140= 42 trees die. The number of trees Tom has left is 140-42= 98 trees. The answer is boxed{ `98` } |
| 2-models | LLM-Blender | Tom starts with 50 trees and plants 10 trees a year, so after 10 years he will have 50+10*10= 150 trees. However, he also chops down 2 trees a year, so after 10 years he will have 150-2*10= 140 trees. After 10 years, 30% of the trees die, so 30% of 140 is 0.3*140= 42 trees die. The number of trees Tom has left is 140-42= 98 trees. The answer is boxed{ `98` } |
| | GaC | Tom plants 10 trees a year and chops down 2 trees a year, so the net increase in trees each year is 10 - 2 = 8 trees. After 10 years, the total number of trees planted is 10 * 10 = 100 trees. The total number of trees chopped down is 2 * 10 = 20 trees. So the total number of trees after 10 years is 50 + 100 - 20 = 130 trees. If 30% of the trees die, then the number of trees that die is 30/100 * 130 = 39 trees. The number of trees left is 130 - 39 = 91 trees. The answer is boxed{ `91` } |
| | UNITE | Tom plants 10 trees a year, so in 10 years he plants 10*10 = 100 trees. He starts with 50 trees, so after planting 100 trees, he has 50+100 = 150 trees. Every year he chops down 2 trees, so in 10 years he chops down 2*10 = 20 trees. So, after 10 years, he has 150-20 = 130 trees. 30% of the trees die, so 30% of 130 trees is 0.3*130 = 39 trees. Therefore, after 30% of the trees die, he has 130-39 = 91 trees left. The answer is boxed{ `91` } |
| 3-models | LLM-Blender | Tom starts with 50 trees and plants 10 trees a year, so after 10 years he will have 50+10*10= 150 trees. However, he also chops down 2 trees a year, so after 10 years he will have 150-2*10= 140 trees. After 10 years, 30% of the trees die, so 30% of 140 is 0.3*140= 42 trees die. The number of trees Tom has left is 140-42= 98 trees. The answer is boxed{ `98` } |
| | UNITE | Tom plants 10 trees a year and chops down 2 trees a year, so he has 10-2=8 more trees each year. After 10 years, he will have 8*10=80 more trees. He starts with 50 trees, so he will have 50+80=130 trees after 10 years. After 10 years, 30% of the trees die, so 30% of 130 trees die, which is 0.3*130=39 trees. Therefore, he will have 130-39=91 trees left. The answer is boxed{ `91` } |

Table 13: Case study of UNITE and other ensembling methods on GSM8K

