# OpenReview forum: "Determine-Then-Ensemble: Necessity of Top-k Union for Large Language Model Ensembling"
_ICLR.cc/2025/Conference — ICLR 2025 Spotlight_

### Official Review · Reviewer_22sv · 2024-11-02

**Soundness:** 3
**Presentation:** 3
**Contribution:** 3
**Rating:** 8
**Confidence:** 3

**Summary:**

Paper has two main contributions:

1. Presents a thorough analysis of features affecting ensemble performance: bade model performance gap, vocabulary size, and response style consistency.
2. A new ensembling method, UNITE, that focuses on combining only the top-k tokens instead of the entire vocabulary. The method is efficient in runtime and shows a high performance across multiple benchmarks.

**Strengths:**

1. UNITE’s token-level aggregation without full vocabulary alignment is an innovative method which reduced computational needs and making ensembling more efficient.
2. Empirical results are shown over a concrete set of benchmarks.
3. Their focus on model selection strategy brings up insightful practical guidelines which are very useful in practice.
4. They present clear results comparing latency against existing methods.

**Weaknesses:**

1. Lack of analysis on possible limitations or settings or benchmarks where topk-k token alignment fails to improve base models perf.
2. Figures and tables could have more detailed captions with information to be self-contained. (ex figures 4,5 and 6)
3. Figures and plots' font size are very small. Consider increasing the font size to assists readers.

**Questions:**

1. Do you have an insight about why increasing k beyond 10 does not improve perf? I think this result is counter-intuitive, and needs more analysis.
2. What is the impact of base model perf difference when using UNITE?

---

> ### Author Response · Authors · 2024-11-20
> **Author Response to Reviewer 22sv**
>
> We sincerely appreciate the time and effort you put into reviewing our paper. We appreciate the insightful feedback and address your concerns as follows:
>
> ---
>
> **R4W1:** Possible limitations
>
> **R4A1:** We acknowledge that UniTE still faces challenges in the scenarios discussed in Section 3.1, "Impact of Model Performance Discrepancy." As shown in the first column of Table 3, UniTE's performance (73.31) is slightly lower than that of the superior base model (73.46) when the performance gap between the two base models, Mistral and OpenChat, exceeds 10%. However, UniTE exhibits the smallest decrease in performance compared to its competitors.
>
> **R4W2 & R4W3:** Captions and font size
>
> **R4A2**: We apologize for any inconvenience. For Fig. 4, we update the caption to: "Impact of the hyperparameter k on the ARC and TriviaQA datasets. Increasing k beyond a certain point leads to a slight decline or no improvement in performance." For Fig. 6, we revise the caption to: "Comparison of different decoding methods. The greedy decoding strategy is more effective for eliciting the next token in deterministic tasks."  The revised contexts are shown in red in our revised PDF.
>
> **R4Q1:** Why increasing k beyond a range may lead to a slight decline or no improvement?
>
> **R4A3:** Aligning with top-k and top-p sampling, in Fig. 3, we present the token distribution during the generation process, revealing that only a few tokens significantly contribute to the overall probability across the vocabulary. This observation motivates our proposed UniTE approach. Besides, we would like to clarify that the enhanced efficiency and effectiveness stem from the reduced token options and our specialized union mapping method, as outlined in Section 4.2. UniTE constructs a union of the top-k tokens from each model and expands this set using each model’s tokenizer. This is followed by probability aggregation to determine the next token. UniTE avoids the need for auxiliary mapping matrices and full vocabulary alignment, respecting the unique tokenization of each base LLM.
>
> Here we offer a particular instance for clearer demonstration. We make the hypothesis that model 1 and model 2 are heterogeneous and employ different tokenization strategies. Specifically, if V1_[0-10)=[‘Jam’ (0.55), ‘James’ (0.2), ‘Jan’ (0.15) ...], V1_[10-15) = [‘J’ (0.04),....];  V2_[0-10)=[‘Jan’ (0.6), ‘Jam’ (0.21), ‘Ja’ (0.1),...], V2_[10-15) = [‘Janet’ (0.03),....], the result of top-10 union is Vu_10=[‘Jam’ (0.38), ‘Jan’ (0.375),...,]. Suppose ‘Jam’ is the expected token, then the top-10 union with greedy decoding elicits the correct answer. However, if ‘Janet’ does not exist in voucabualry1 then the tokenizer1 turns ‘Janet’ into ‘Jan’ and ‘et’, the top-15 tokens union changes into: Vu_15=[‘Jan’ (0.39), ‘Jam’ (0.38),...,] thus the next token is wrong ‘Jan’.
>
> **R4Q2:** The impact of base model performance when using UniTE
>
> **R4A4:**  We would like to note that in Fig. 1 (Section 3.1), we also illustrate the impact of performance disparity among models using UniTE. UniTE demonstrates performance similar to that of other ensemble methods, as a substantial performance gap results in performance declines.
>
> Besides, we would like to offer more statistics of UniTE related to experiments listed in Fig. 2. We identify model pairs based on the GSM8K dataset with performance gaps of approximately 40% (LLaMA2-7BChat and Mistral-7B-Instruct-v0.3), 25% (LLaMA2-13b-Chat and Mistral-7B-Instruct-v0.3), 15% (OpenChat-3.5 and Mistral-7B-Instruct-v0.3), and less than 10% (LLaMA3-8B-Instruct and Qwen2- 7B-Instruct). The results are shown below:
>
> |                      | GSM(40%) |                      | GSM(25%) |                       | GSM(15%) |                     | GSM(Similar) |
> |----------------------|----------|----------------------|----------|-----------------------|----------|---------------------|--------------|
> | LLaMA2-7b            | 17.66    | LLaMA2-13b           | 31.77    | OpenChat-7b           | 73.46    | LLaMA3-8b           | 78.77        |
> | Mistral-7b           | 56.48    | Mistral-7b           | 56.48    | Mistral-7b            | 56.48    | Qwen2-7b            | 80.97        |
> | UniTE (base LLaMA2)  | 34.67    | UniTE (base LLaMA2)  | 50.12    | UniTE (base OpenChat) | 73.16    | UniTE (base LLaMA3) | 82.71        |
> | UniTE (base Mistral) | 51.33    | UniTE (base Mistral) | 55.67    | UniTE (base Mistral)  | 57.33    | UniTE (base Qwen2)  | 84.99        |
>
> The table indicates that as the performance gap increases, the benefits of ensembling inferior models become more pronounced. When the performance difference is within 10%, ensembling can lead to improved results. Additionally, we emphasize that UniTE exhibits a smaller decrease in performance compared to the other methods shown in Fig. 2, further validating its effectiveness.
>
> ---
>
>
> We sincerely thank the reviewer again for the helpful feedback!

---

> > ### Comment · Reviewer_22sv · 2024-11-24
> >
> > Thank you for the clarifications, and great work! I choose to keep my score.

---

### Official Review · Reviewer_QXTA · 2024-11-04

**Soundness:** 3
**Presentation:** 3
**Contribution:** 3
**Rating:** 8
**Confidence:** 3

**Summary:**

This paper proposed an LLM ensemble method, which is to ensemble among only top-k tokens probability.

**Strengths:**

1. It includes good analysis and preliminary experiments for LLM ensembling.
2. Based on the preliminary analysis, they reduce the vocabulary for the previous method into the top-k selection for the model ensembling.

**Weaknesses:**

1. I believe the experiments should include the efficiency compared to the previous approach.
2. It seems, in theory, that including the whole vocabulary should work better, although maybe marginally or at least the same compared to top-k, since it includes the whole picture of the token distribution. I am questioning your experimental results because top-k is better in every dataset, which logically cannot be the case.

**Questions:**

Like Weaknesses#2 above, can you provide reasons or analysis why the top-k approach is better in performance than the whole vocabulary on performance?

---

> ### Author Response · Authors · 2024-11-20
> **Author Response to Reviewer QXTA (1)**
>
> We appreciate the valuable advice from the reviewer. And we would like to address your concerns as follows:
>
> ---
>
> **R3W1:** Efficiency compared to the previous approach.
>
> **R3A1:**  Fig. 5 in “Section 5.4 Further Analysis” illustrates the latency of various methods. The results indicate that the latencies for the individual Mistral and OpenChat models are 73.05 ms/token and 75.09 ms/token, respectively, under our hardware device settings, while DeePen and GaC exhibit latencies of 225.13 ms/token and 122.73 ms/token. Notably, UniTE's latency is **87.78 ms/token**, which is significantly lower than that of previous ensemble methods and only 16.8% higher than that of the single base model. Regarding memory consumption, as outlined in the introduction and Table 5, UniTE utilizes only **0.04%** of the vocabulary tokens, with memory usage primarily related to model deployment—a necessity common to all ensemble learning approaches. In contrast, DeePen and GaC require additional memory to store their extensive intersection and union vocabularies.
>
> **R3W2 & R3Q1:** Why does UniTE with Top-k ensembling achieve better performance?
>
> **R3A2:**  First, the top-k ensembling is motivated by the decoding strategies, like top-k sampling[1] and top-p sampling[2] in the field of neural text generation, which suggest that in each generation step, only a few tokens significantly contribute to the overall probability across the vocabulary. **Therefore, we clarify that the enhanced efficiency and effectiveness arise from reduced token options and our specialized union mapping method, as outlined in Section 4.2. UniTE constructs a union of the top-k tokens from each model and expands this set using each model’s tokenizer, followed by probability aggregation to determine the next token.** UniTE only focuses on the important tokens and eliminates the noise from other irrelevant tokens, and also respects the unique tokenization of each base LLM.
>
> We also highlight the distinctions between UniTE and the methods of DeePen [3] and GaC [4], which utilize the entire vocabulary for alignment. DeePen selects the intersection of base models as anchor words and employs embeddings to map other vocabulary items into a common space, relying on these anchor tokens instead of the full vocabulary. To ensure effective representation, DeePen includes all tokens from the intersection, as a larger subset is advantageous. GaC projects the probability vectors of multiple LLMs into a unified vocabulary dimension using a mapping matrix, aggregating outputs at each generation step to select the next token. In contrast, UniTE avoids the complexities of embedding mappings and adopts a novel approach to top-k union construction.
>
> We further test k with extremely large values to mimic aligning on the whole vocabulary with our method. As shown in Table below, further increasing k leads to either a slight decline or no change in performance. This finding reinforces our assertion that, in probability-level ensembling, it is unnecessary to align the entire vocabulary to predict the next token.
>
>
> |          | TriviaQA |
> |----------|----------|
> | Mistral  | 64.30    |
> | OpenChat | 61.77    |
> | K=5      | 64.52    |
> | K=10     | 65.80    |
> | K=20     | 65.77    |
> | K=100    | 65.65    |
> | K=1000   | 65.74    |
> | K=10000  | 65.72    |
>
> Here we offer a particular instance for clearer demonstration. We make the hypothesis that model 1 and model 2 are heterogeneous and employ different tokenization strategies. Specifically, if V1_[0-10)=[‘Jam’ (0.55), ‘James’ (0.2), ‘Jan’ (0.15) ...], V1_[10-15) = [‘J’ (0.04),....];  V2_[0-10)=[‘Jan’ (0.6), ‘Jam’ (0.21), ‘Ja’ (0.1),...], V2_[10-15) = [‘Janet’ (0.03),....], the result of top-10 union is Vu_10=[‘Jam’ (0.38), ‘Jan’ (0.375),...,]. Suppose ‘Jam’ is the expected token, then the top-10 union with greedy decoding elicits the correct answer. However, if ‘Janet’ does not exist in voucabualry1 then the tokenizer1 turns ‘Janet’ into ‘Jan’ and ‘et’, the top-15 tokens union changes into: Vu_15=[‘Jan’ (0.39), ‘Jam’ (0.38),...,] thus the next token is wrong ‘Jan’.
>
> Moreover, to alleviate your concern about our experimental results, we have uploaded our code to supplementary materials.
>
> ---
>
> We sincerely thank the reviewer again for the helpful feedback!
>
> References:
>
> [1] Hierarchical Neural Story Generation. ACL 2018.
>
> [2] The Curious Case of Neural Text Degeneration. ICLR 2020.
>
> [3] Ensemble Learning for Heterogeneous Large Language Models with Deep Parallel Collaboration. NeurIPS 2024.
>
> [4] Breaking the Ceiling of the LLM Community by Treating Token Generation as a Classification for Ensembling. EMNLP 2024

---

> > ### Comment · Reviewer_QXTA · 2024-11-26
> >
> > Thanks. The author addressed my concern, and I increased the score.

---

> ### Author Response · Authors · 2024-11-25
> **Further discussions are appreciated!**
>
> Dear Reviewer QXTA,
>
> Thank you sincerely for taking the time to review our submission and for your insightful comments. We have carefully considered your concerns and have addressed them in our recent responses. Your constructive feedback has been instrumental in helping us improve the quality of our work.
>
> As the deadline for the discussion period approaches, we would greatly appreciate it if you could review our responses and let us know whether they adequately address your concerns. If any issues remain unresolved, please share your remaining concerns so that we can respond appropriately before the deadline. We welcome any further discussions you may wish to have.
>
> If you find that our responses satisfactorily address your concerns, we would be grateful if you could consider increasing the rating score.
>
> We understand how busy you are, and we truly appreciate your time and effort. We look forward to your further comments and discussions.
>
> Best wishes,
>
> Authors

---

### Official Review · Reviewer_qMEZ · 2024-11-04

**Soundness:** 3
**Presentation:** 3
**Contribution:** 3
**Rating:** 8
**Confidence:** 3

**Summary:**

The paper identifies and tests three hypotheses about importance factors that influence the performance of logit-based LLM ensembles: performance discrepancy, vocabulary size differences, and stylistic response differences, and provides guidelines for choosing models to ensemble with. The authors propose UNITE, an ensembling method that uses the top-k logits from each model, and show that it outperforms numerous ensembling baselines while being significantly cheaper.

**Strengths:**

- The paper presents concrete empirical conclusions about what factors are actually important for ensembling (performance gap, response style) or not (vocabulary size difference.)
- It presents UNITE, a novel top-k union ensembling approach that drops the computational overhead tremendously by using the top-k tokens of each model instead of the full vocabulary, and that obviates the need for full vocabulary alignment.
- UNITE significantly outperforms the baselines (LLM-Blender, DeePen, GAC) in average accuracy across six popular benchmarks for question-answering, reasoning, and knowledge.
- The paper is clear and well written.

**Weaknesses:**

- The analysis in Conclusion II on vocabulary size differences is restricted to English language tasks. It may benefit from a discussion of multilinguality, where tokenization might be less consistent between models.
- In Conclusion III the paper identifies differences in response style as a major problem for ensembling models. The proposed solution of limiting longer responses to 2x the length of shorter ones would benefit from theoretical justification. A robust solution to this problem which enables ensembling of models with different response styles would be ideal, as the current approach limits the practicality of the method.

**Questions:**

Does the optimal value of k for the top-k tokens vary across tasks and domains? Is there a principled way to determine k?

---

> ### Author Response · Authors · 2024-11-20
> **Author Response to Reviewer qMEZ (1)**
>
> We greatly appreciate the reviewer’s helpful feedback. Below, we address your concerns:
>
> ---
>
> **R2W1:** Whether Conclusion II works in another language.
>
> **R2A1:**  We also conducted experiments on the CMMLU [1], a Chinese Multitask Language Understanding Evaluation benchmark to validate our findings. We used Yi-6B (vocabulary size: 64,000) and Qwen2-7b-instruct (vocabulary size: 152,064) as our base models, both supporting the Chinese language. Qwen2-7b-instruct is the primary model, the results of various ensemble approaches are presented below:
>
> |                   | CMMLU |
> |-------------------|-------|
> | Yi-6B             | 75.21 |
> | Qwen2-7B-instruct | 83.22 |
> | LLM-Blender       | 79.08 |
> | DeePen            | oom   |
> | GaC               | 75.88 |
> | UniTE             | 83.89 |
>
> Similar to the results listed in Section 3.2, irrespective of the gap in vocabulary size, existing methods still demonstrate improvements, thereby indicating that vocabulary size for model ensembling is marginal. We have included this experiment in our revised PDF version.
>
> **R2W2:** Solution to different response styles.
>
> **R2A2:** We would like to adopt preprocessing steps to cope with different response styles. In our preliminary experiments, we tested several datasets and observed significant differences in response styles for TriviaQA when using Qwen compared to LLaMA series models as shown in Table 2. For instance, following the original 5-shot prompt settings of previous work[4][5], Qwen2.5 analyzes and includes conclusions in its responses, complicating the extraction of solution-oriented knowledge for QA tasks, while LLaMA3 provides the solution directly. To force responses from different models in a similar style and avoid Qwen2.5 responding with verbose analysis, we employed a new 5-shot prompt designed to elicit answers in the format "The answer is xxx.".  The responses are presented in the table below.
>
> | TriviaQA Question                                     | Which Lloyd Webber musical premiered in the US on 10th December 1993?                                                                                                                    |
> |-------------------------------------------------------|------------------------------------------------------------------------------------------------------------------------------------------------------------------------------------------|
> | Original prompt (Response style referring to Table 2) | Question: In the 1971 Number One hit Ernie by Benny Hill, what was the name of Ernie's horse who was kicked by his rival, Two-ton Ted from Teddington? Answer: **Triggers**.           |
> | New Prompt                                            | Question: In the 1971 Number One hit Ernie by Benny Hill, what was the name of Ernie's horse who was kicked by his rival, Two-ton Ted from Teddington? Answer: **The answer is Triggers**. |
> | LLaMA3 response                                       | The answer is Sunset Boulevard.                                                                                                                                                          |
> | Qwen2.5 response                                      | The answer is Sunset Boulevard.                                                                                                                                                          |
>
> Then we test different ensemble approaches:
>
> |             | TriviaQA   |
> |-------------|------------|
> | LLaMA3      | 70.68 (67) |
> | Qwen2.5     | 57.85 (52) |
> | LLM-Blender | 64.77      |
> | UniTE       | 67.45      |
>
> The original prompts elicit accuracy in the brackets (As the tedious response style illustrated in Table 2, Qwen incorporates answers into the analysis, we randomly sampled 100 instances from the 1500-test set to manually extract the predictions). After adjusting the prompt, we can easily analyze the results for the entire test set. Consistent with our findings presented in the main text, when the base model’s performance gap exceeds 10%, ensemble learning may yield little to no improvement. Additionally, it is important to note that our UniTE approach still outperforms its competitors.

---

> > ### Author Response · Authors · 2024-11-20
> > **Author Response to Reviewer qMEZ (2)**
> >
> > **R2Q1:** Selection of hyperparameter k.
> >
> > **R2A3:**  Align with top-k[2] and top-p sampling[3], in Fig. 3, we present the token distribution during the generation process, revealing that only a few tokens significantly contribute to the overall probability across the vocabulary. This observation motivates our proposed UniTE approach. In section 5.3 “Ablation Study”, we further discuss the effect of hyperparameter k. We conduct experiments using Mistral and OpenChat models on the TriviaQA and ARC-C datasets. As illustrated in Fig. 4, increasing k up to 10 enhances performance significantly. However, further increasing k beyond 10 leads to either a slight decline or no change in performance. Hereby, we suggest a general selection of k as 10.
> >
> > ---
> >
> >
> > Again, we sincerely thank you for the valuable suggestions!
> >
> > References:
> >
> > [1] CMMLU: Measuring massive multitask language understanding in Chinese. ACL(findings) 2024.
> >
> > [2] Hierarchical Neural Story Generation. ACL 2018.
> >
> > [3] The Curious Case of Neural Text Degeneration. ICLR 2020.
> >
> > [4] TriviaQA: A Large Scale Distantly Supervised Challenge Dataset for Reading Comprehension. ACL 2017
> >
> > [5] Ensemble Learning for Heterogeneous Large Language Models with Deep Parallel Collaboration. NeurIPS 2024.

---

> > > ### Comment · Reviewer_qMEZ · 2024-11-24
> > >
> > > Thank you very much for the thoughtful replies! I will keep the score.

---

### Official Review · Reviewer_nt9F · 2024-11-05

**Soundness:** 2
**Presentation:** 3
**Contribution:** 3
**Rating:** 6
**Confidence:** 4

**Summary:**

This paper introduces a novel ensembling approach, UNITE (Union Top-k Ensembling), that efficiently integrates large language models (LLMs) by focusing on the union of top-k tokens from each model rather than aligning the full vocabulary. It seeks to improve the computational efficiency and effectiveness of LLM ensembles by addressing key issues of compatibility, vocabulary size, and response styles. The authors propose a model selection strategy to identify compatible models, limiting the influence of incompatible LLMs on the ensemble's performance. Experimental results across multiple benchmarks validate the benefits of UNITE in enhancing performance, reducing latency, and decreasing the computational burden.

**Strengths:**

- The paper identifies compatibility challenges in LLM ensembling and focuses on top-k tokens, aligning this strategy with empirical evidence that vocabulary alignment often introduces computational inefficiencies.
- The authors conduct extensive experiments on multiple models and benchmarks, analyzing factors like vocabulary size, response style, and latency. The results support UNITE’s superiority in maintaining high performance while minimizing latency and token manipulation.
-  The proposed determine-then-ensemble strategy offers a generalizable framework for selecting compatible models, making the findings applicable to real-world LLM applications that require efficient model collaboration.

**Weaknesses:**

- While the paper addresses task-specific challenges, it could benefit from deeper exploration into why certain tasks (like MMLU) see greater performance improvements than others. Further insight into how task characteristics impact ensembling effectiveness would add depth to the analysis.
- The model selection process relies on response style consistency and performance alignment within a 10% range, which may limit scalability when dealing with a large pool of candidate models. The method would benefit from a more automated or quantitative metric for determining compatibility.
- While UNITE is evaluated across standard datasets, some benchmarks like GSM8K and TriviaQA may not fully capture the diverse range of LLM applications. Including more varied tasks could strengthen the argument for UNITE’s general applicability.

**Questions:**

- Could you provide more theoretical support or intuition for why limiting alignment to the top-k tokens effectively enhances ensemble performance? How does this approach balance between accuracy and computational efficiency at a probabilistic level?
- How does UNITE handle models with markedly different response styles in practice? Would introducing a preprocessing step to standardize response formats (e.g., for tasks like QA or summarization) enhance compatibility?
- Since UNITE is partially motivated by efficiency, adding a comparative breakdown of memory and latency for each method would clarify the computational trade-offs involved. Including charts or tables that detail average and worst-case latency per token would help underscore UNITE’s operational benefits.

---

> ### Author Response · Authors · 2024-11-20
> **Author Response to Reviewer nt9F (1)**
>
> We sincerely thank the reviewer for the constructive suggestions, which help us to improve the quality of our work, and are pleased that you find our work to be novel and effective. We would like to address your concerns as follows:
>
> ---
> **R1W1:** Why does UniTE achieve greater improvements on the MMLU dataset?
>
> **R1A1:** MMLU is a benchmark comprising many subject-specific tasks (e.g. physics, biology, chemistry). We randomly selected five subjects from this collection and conducted experiments using LLaMA3-8B-Instruct and Qwen2-7B-Instruct, with the results presented in the table below. Notably, although the performance of LLaMA3 and Qwen2 is comparable across the entire MMLU dataset, significant differences emerge in their performance on individual subsets. This suggests that LLaMA3 and Qwen2 exhibit substantial differences in their capabilities across various subjects, making them complementary to each other on the MMLU benchmark.
>
> |                        | Qwen2-7b-instruct | LLaMA3-8b-instruct |
> |------------------------|-------------------|--------------------|
> | high_school_us_history | 84.31             | 80.39              |
> | security_studies       | 73.88             | 74.26              |
> | abstrct_algebra        | 48                | 36                 |
> | conceptual_physics     | 71.06             | 57.02              |
> | logical_fallacies      | 76.69             | 74.85              |
> | **Overall MMLU**           | 64.96             | 64.58              |
>
> Assuming we choose LLaMA3 as the base model, when LLaMA3's performance on a subset is inferior to that of Qwen2, the ensembling is likely to yield a more substantial enhancement. Therefore, the application of UniTE results in a more pronounced improvement across the entire MMLU dataset.
>
> **R1W2:** Scalability of our methods regarding performance gap limitation.
>
> **R1A2:** As our first key takeaway says that “smaller performance gaps facilitate greater gains from model ensembling”, we recommend to ensemble the comparative models on the target tasks to obtain better performance than applying the single base model. We would like to clarify that the performance gap within 10% is not the hard limitation of our method. UniTE can definitely be applied to all candidate model pairs, and can consistently perform effectively, regardless of the performance gap, if the intention is merely to enhance the inferior model using the superior one. However, if users aim for performance that even surpasses the superior model, this factor should be considered. We believe the performance alignment for optimal model ensembling is both intuitive and reasonable since a significantly weaker model is unlikely to contribute valuable information to a stronger model. Our extensive experiments also confirm this view.
>
> As for how to automatically deal with a large pool of candidate models, we detailed the base model selection pipeline in Section 4.1. Firstly, we choose the best-performing model for the target task. Subsequently, select the next best-performing model that satisfies the criteria (performance alignment and response style consistency) for successful ensembling with the first chosen model, continuing this process iteratively until the maximum number of base models is reached or no further suitable models can be found. We also provide an alternative way to alleviate the examination of response style consistency in the later part. So, the entire selection process can be automated when the performance of candidate models on the target task is accessible. In cases where these performance scores are not explicitly available, we recommend to sample a calibration dataset from the target task and obtain a reference score for each model, which can then be used to assess compatibility.
>
> **R1W3:** Task choice menus
>
> **R1A3：** To validate the performance of UniTE and ensure a fair comparison with previous works [1][2], we have already evaluated three main categories: 1) Comprehensive examination (MMLU, ARC-C), 2) Reasoning capabilities (GSM8K, PIQA), and 3) Knowledge capacities (TriviaQA, NQ).
>
> We also conduct additional experiments to address the reviewer's concerns using the BBH (BIG-Bench Hard) benchmark [3],  a diverse evaluation suite of 23 challenging tasks such as symbolic reasoning. Due to computational constraints and limited rebuttal time, we randomly evaluated 10 subsets, and the results are presented below. Consistent with our main findings, UniTE demonstrates superior performance compared to other methods, highlighting the effectiveness and generalizability of our approach. We include these results in our revised PDF version in red.
>
>
> |                    | BBH   |
> |--------------------|-------|
> | LLaMA3-8b-instruct | 73.00 |
> | Qwen2-7b-instruct  | 68.60 |
> | LLM-Blender        | 68.79 |
> | DeePen             | oom   |
> | GAC                | 69.86 |
> | UniTE              | 73.52 |

---

> > ### Author Response · Authors · 2024-11-20
> > **Author Response to Reviewer nt9F (2)**
> >
> > **R1Q1:** Why does limiting alignment to the top-k tokens effectively enhance ensemble performance? How does this approach balance between accuracy and computational efficiency?
> >
> > **R1A4:** In Fig. 3, we present the token distribution during the generation process, revealing that only a few tokens significantly contribute to the overall probability across the vocabulary. This observation motivates our proposed UniTE approach. **We would like to emphasize that the enhanced efficiency and effectiveness stem from the reduced token options as well as our specialized union mapping criteria, as outlined in Section 4.2. UniTE constructs a union of the top-k tokens from each model and expands this set using each model’s tokenizer.** This is followed by probability aggregation to determine the next token. UniTE avoids the need for auxiliary mapping matrices and full vocabulary alignment, respecting the unique tokenization of each base LLM.
> >
> > Regarding the balance issue, as illustrated in Fig. 4, we evaluate different k values across different tasks. Increasing k from 5 to 10 results in significant performance improvements; however, further increases beyond 10 do not yield better results and impose additional computational burdens. Therefore, we recommend setting k to 10 in the implementation.
> >
> > **R1Q2:** Handle different response styles.
> >
> > **R1A5:** According to the reviewer’s advice, we try to address the response style issue via preprocessing steps. Specifically, we provide an alternative simple solution by using the few-shot examples to standardize the response format.
> >
> > In our preliminary experiments, we tested several datasets and observed significant differences in response styles for TriviaQA when using Qwen compared to LLaMA series models as shown in Table 2. For instance, following the original 5-shot prompt settings of previous work[1][4], Qwen2.5 analyzes and includes conclusions in its responses, complicating the extraction of solution-oriented knowledge for QA tasks, while LLaMA3 provides the solution directly. To force responses from different models in a similar style and avoid Qwen2.5 responding with verbose analysis, we employed a new 5-shot prompt designed to elicit answers in the format "The answer is xxx.".  The responses are presented in the table below.
> >
> > | TriviaQA Question                                     | Which Lloyd Webber musical premiered in the US on 10th December 1993?                                                                                                                    |
> > |-------------------------------------------------------|------------------------------------------------------------------------------------------------------------------------------------------------------------------------------------------|
> > | Original prompt (Response style referring to Table 2) | Question: In the 1971 Number One hit Ernie by Benny Hill, what was the name of Ernie's horse who was kicked by his rival, Two-ton Ted from Teddington? Answer: **Triggers**.           |
> > | New Prompt                                            | Question: In the 1971 Number One hit Ernie by Benny Hill, what was the name of Ernie's horse who was kicked by his rival, Two-ton Ted from Teddington? Answer: **The answer is Triggers**. |
> > | LLaMA3 response                                       | The answer is Sunset Boulevard.                                                                                                                                                          |
> > | Qwen2.5 response                                      | The answer is Sunset Boulevard.                                                                                                                                                          |
> >
> > Then we test different ensemble approaches:
> >
> > |             | TriviaQA   |
> > |-------------|------------|
> > | LLaMA3      | 70.68 (67) |
> > | Qwen2.5     | 57.85 (52) |
> > | LLM-Blender | 64.77      |
> > | UniTE       | 67.45      |
> >
> > The original prompts elicit accuracy in the brackets (As the tedious response style illustrated in Table 2, Qwen incorporates answers into the analysis, we randomly sampled 100 instances from the 1500-test set to manually extract the predictions). After adjusting the prompt, we can easily analyze the results for the entire test set. Consistent with our findings presented in the main text, when the base model’s performance gap exceeds 10%, ensemble learning may yield little to no improvement. Additionally, it is important to note that our UniTE approach still outperforms its competitors.

---

> > > ### Author Response · Authors · 2024-11-20
> > > **Author Response to Reviewer nt9F (3)**
> > >
> > > **R1Q3:** Latency and memory analysis.
> > >
> > > **R1A6:**  We would like to clarify that Fig. 5 illustrates the latency of various methods. The results indicate that the latencies for the individual Mistral and OpenChat models are 73.05 ms/token and 75.09 ms/token, respectively, under our hardware device settings, while DeePen and GaC exhibit latencies of 225.13 ms/token and 122.73 ms/token. Notably, UniTE's latency is **87.78 ms/token** , which is significantly lower than that of previous ensemble methods and only 16.8% higher than that of the single base model.
> > >
> > > Regarding memory consumption, as outlined in the Introduction and Table 5, UniTE utilizes only **0.04%** of tokens of the whole vocabulary, with memory usage primarily related to model deployment—a necessity common to all ensemble learning approaches. In contrast, DeePen and GaC require additional memory to store their extensive intersection and union vocabularies.
> > >
> > > ---
> > >
> > > Again, we sincerely thank the reviewer for the valuable suggestions!
> > >
> > > Reference:
> > >
> > > [1] Ensemble Learning for Heterogeneous Large Language Models with Deep Parallel Collaboration. NeurIPS 2024.
> > >
> > > [2] Breaking the Ceiling of the LLM Community by Treating Token Generation as a Classification for Ensembling. EMNLP 2024
> > >
> > > [3] Challenging BIG-Bench Tasks and Whether Chain-of-Thought Can Solve Them. ACL 2023
> > >
> > > [4] TriviaQA: A Large Scale Distantly Supervised Challenge Dataset for Reading Comprehension. ACL 2017

---

> ### Author Response · Authors · 2024-11-25
> **Further discussions are appreciated!**
>
> Dear Reviewer nt9F,
>
> We sincerely appreciate your time to review our submission and provide valuable comments. We have carefully considered your concerns and tried to resolve them in our rebuttal. Your constructive feedback will greatly help us improve the quality of the work.
>
> As the deadline of discussion period is apporaching, we would really appreciate it if you have read our response and let us konw whether the previous responses have addressed your concerns accordingly. If your concerns have not been well resolved, could you please let us know your remaining concerns so that we have the opportunity to respond before the deadline? We are happy to have any follow-up discussions. If you are satisfied with our response and it truly addresses your concerns, we would really appeciate it if you could consider to increase the rating score.
>
> We understand you are very busy and we really appreciate your time. Looking forward to your further comments and discussions.
>
> Best wishes,
>
> Authors

---

> > ### Comment · Reviewer_nt9F · 2024-11-27
> >
> > Thank you for your effort in addressing my concerns. Your response was quite helpful, but I want to keep my score.

---

### Meta-Review · Area_Chair_akW1 · 2024-12-18

**Metareview:**

Many LLMs have been released recently, each of which is trained on different data and has its strengths and weaknesses when performing downstream tasks. This paper proposes a new method for ensembling LLMs, UNITE (UNIon Top-k Ensembling), to achieve better performance than any individual language model in the ensemble. UNITE ensembles LLMs by taking the union of the top-k tokens predicted by each model rather than considering the full vocabulary of the models; thus, the approach is more efficient than prior LLM ensembling methods and handles the compatibility challenges that come with ensembling diverse LLMs with different vocabularies. Furthermore, the experiments show that UNITE outperforms these prior approaches on multiple downstream tasks, despite the reduced computational overhead of the method.

Strengths:
- UNITE is an effective, general method for ensembling LLMs that improves over prior methods in this space while also solving an existing issue (compatibility across vocabularies)and reducing the computational costs of ensembling (nt9F, qMEZ, 22sv).
- The paper also extensively analyzes the factors that make ensembing LLMs effective (nt9F, qMEZ, QXTA).
- During revisions, the authors incorporated multiple experiments to address the reviewers' concerns, including a new LLM benchmark and an additional evaluation in a new language (Chinese) with CMMLU.

Weaknesses:
While the paper contains evaluations on many different benchmarks, there is limited analysis as to why some benchmarks benefit more from the UNITE method than others (nt9F) or in cases when the method fails (22sv).

Most of the other weaknesses raised by the reviewers, such as the effect of response style, were fully addressed in the author response. I recommend that the authors increase the size of the font in the figures, as they are currently difficult to read.

**Additional Comments On Reviewer Discussion:**

The authors provided comprehensive responses to the reviewers and, in many cases, included new experiments to address concerns and questions; many of these experiments were also added to the paper. In response, multiple reviewers chose to raise their scores.

---

### Decision · Program_Chairs · 2025-01-22

Accept (Spotlight)